

# Mechanical and chemical element structures of sea urchin spines: specialization in ambulacral and interambulacral areas

Pathitta Suteecharuwat[1], Mayuka Arakawa[2] and Yutaka Yoshida[3]

[1] Kitami Institute of Technology, Graduate School of Engineering, Kitami, Hokkaido, Japan
[2] Kitami Institute of Technology, School of Regional Innovation and Social Design Engineering Intelligent Machines and Biomechanics Course Program, Kitami, Hokkaido, Japan
[3] Kitami Institute of Technology, Faculty of Engineering, Kitami, Hokkaido, Japan

Corresponding author
Yutaka Yoshida,
yyoshida@mail.kitami-it.ac.jp

## ABSTRACT

Sea urchin spines are of interest for biomaterials and functional materials development due to their mechanical properties, which depend on their elemental composition. However, no previous study has examined the structural distinctions between the spines in the ambulacral and interambulacral areas. This study addresses that gap by investigating the structural and mechanical differences in the spines of *Strongylocentrotus nudus*, with a focus on these two areas. We used cantilever bending tests, Fourier-transform infrared (FT-IR) spectroscopy, X-ray diffraction (XRD), and inductively coupled plasma atomic emission spectroscopy (ICP-AES) to analyze the composition, elasticity, and microstructure of the spines. The bending modulus of elasticity was higher in the ambulacral area (52.067 GPa) compared to the interambulacral area (10.133 GPa), hardness and deformation. ICP-AES analysis revealed that ambulacral shaft had a slightly higher concentration of magnesium (Mg) (0.9844 wt%) compared to the interambulacral shaft (0.9804 wt%), while the calcium (Ca) concentration was lower in the ambulacral shaft (39.6578 wt%) compared to the interambulacral shaft (42.1076 wt%). Furthermore, a variation in Mg concentration was observed between the base and shaft parts of the spine. XRD showed a narrower (104) lattice spacing in the ambulacral spine (3.0264 Å) compared to the interambulacral spine (3.0275 Å), correlating with higher Mg concentration. These compositional and structural differences suggested that *S. nudus* modulates Mg concentration in calcite to achieve functional specialization of spines for locomotion and defense. Our findings may be useful for the development of novel functional materials.

## INTRODUCTION

Sea urchins are marine echinoderms with spherical or elliptical shapes and numerous spines on their tests (*Smith & Kroh, 2006*; *Lawrence, 2013*). Sea urchin shells and spines are composed of a magnesium (Mg) and calcium (Ca) carbonate structure

(*Vecchio et al., 2007*; *Moureaux et al., 2010*; *Albéric et al., 2019*). The spines of *Strongylocentrotus nudus* are composed of stereom structure, a porous, mesh-like microstructure characterized by numerous internal cavities, which make them both lightweight and strong (*Moureaux et al., 2010*; *Gorzelak et al., 2011*; *Albéric et al., 2019*). The mechanical properties of spines depend partly on their constituent elements, such as strength, hardness, and elasticity (*Tsafnat et al., 2012*; *Lauer et al., 2020*; *Cölfen et al., 2022*). These spine structures are expected to be useful in biomaterials and functional materials development because they are optimal for locomotion and defense (*Voulgaris et al., 2021*; *Emerson et al., 2017*). The sea urchin test (skeleton) is divided into ambulacral and interambulacral areas. The ambulacral areas include pores for tube feet used in locomotion, while interambulacral areas are located between them and serve primarily as rigid structural support (*Gao et al., 2015*). Earlier work on the microstructure of echinoid stereom by *Smith (1980)* established the foundational understanding of stereom morphology and function. Additionally, pioneering research by *Weber (1969)* and *Weber et al. (1969)* provided crucial insights into the chemical composition and unusual strength properties of echinoderm calcite, which are highly relevant to the present study.

The bending behavior of the cantilever is crucial for locomotion as it serves as an indicator of the strength and flexibility of the sea urchin spine. Herein, we examined the relationship between mechanical properties and microstructure using a cantilever bending test and other exact analyses, which indicated that sea urchins control the Mg concentrations to acquire these functions.

In *S. nudus*, spines in the ambulacral area tend to be shorter and thicker. In contrast, those in the interambulacral area are generally longer and thinner, reflecting their distinct roles in locomotion and defense. Tube feet located in the ambulacral area facilitate locomotion across the substrate. During locomotion, the spines support the body by providing mechanical stability and balance (*Santos et al., 2005*). The longer interambulacral spines can radiate outward in response to stimuli, forming a physical barrier that helps deter predators. While the spines respond defensively to touch, locomotion is controlled by the tube feet (*Yu et al., 2019*; *Voulgaris et al., 2021*; *Thompson et al., 2021*; *Hebert, Silvia & Wessel, 2024*). Currently, no study has elucidated the structural distinctions between the spines in the ambulacral and interambulacral areas, which are involved in locomotion and defense in sea urchins. Additionally, these spines may have functions other than locomotion and defense. We hypothesized that the two spine types have different functional characteristics. We thus collected the spines of the sea urchin, *S. nudus* and examined them from the ambulacral and interambulacral areas based on their bending properties, crystalline structure, and Mg concentrations. In addition, the structural details were investigated using Fourier-transform infrared (FT-IR) spectroscopy, X-ray diffraction (XRD), and inductively coupled plasma atomic emission spectroscopy (ICP-AES). These structural analyses were performed to determine the relationship between the mechanical properties and constituent elements (Mg and Ca). *S. nudus* was selected due to its abundance in northern Japan and its distinct spine differentiation,

making it suitable for comparative studies in echinoid biomechanics. In this study, we focused only on primary spines from the ambital region of the corona to minimize variation due to positional differences along the oral–aboral axis.

## MATERIALS AND METHODS

### Sample preparation

Spines of adult sea urchins (*S. nudus*) were procured along with shells from Rishiri Island, Hokkaido Prefecture, Japan (approximate coordinates: 45.0700°N, 141.2380°E). Although the exact depth was not recorded during collection, the specimens were obtained from intertidal to shallow subtidal zones, which are typically within 1–3 m depth. While the total number of individuals was not recorded, spines were randomly selected from a pool of multiple individuals based on morphological integrity, ensuring that only complete, undamaged spines were used for analysis. Spines were visually screened to exclude regenerated specimens with blunt ends or abnormal internal structure. The shells were thoroughly washed, and the organic tissue was removed before being dehydrated at room temperature (25 °C). The sea urchin spines were then extracted from the shell by dividing them into ambulacral and interambulacral areas (Figs. 1 and 2) and stored in desiccators at $10^{-2}$ Pa. To remove adhering soft tissue, the shells with attached spines were immersed in distilled water to soften organic residues. The spines were then brushed gently but thoroughly to clean the external surfaces without applying any chemical or enzymatic treatments. After drying in ambient conditions and further drying under reduced pressure in a vacuum chamber, specimens were stored in a desiccator until analysis. Only primary spines located in the ambital region were used for all analysis in this study. Secondary spines, which are morphologically smaller and more variable, were excluded to reduce functional and structural variation. Table 1 shows the average spine dimensions, including total length, base length, shaft length, and diameter of the spine. The analyzed specimens are deposited in the Faculty of Engineering, Kitami Institute of Technology, Japan.

### Cantilever bending test

For the cantilever bending tests, the base of the sea urchin spine was embedded in an aluminum pipe using ultraviolet-hardening acrylate resin. To ensure consistent loading conditions, all spines were oriented such that the indenter applied force perpendicular to the shaft, 1 mm below the tip, thereby minimizing the effect of minor asymmetries in spine morphology. To minimize the influence of spine shape variation, all specimens were indented at a uniform position 1 mm from the tip. The sea urchin spine in the ambulacral and interambulacral areas was adjusted to an indenter position 1 mm from the spine tip and fixed using a jig for each test (Fig. 3). The loading force and displacement were recorded using an autograph (MX2-500N, ZTA-20N; Imada, Aichi, Japan). In the bending test, 10 spines were used per area. The indenter of a load capability of 20 kN was applied at a 10 mm/min speed. Dimensions such as the total length, base and shaft length, and load point diameter of the sea urchin spines were measured pre- and post-testing. Fractured
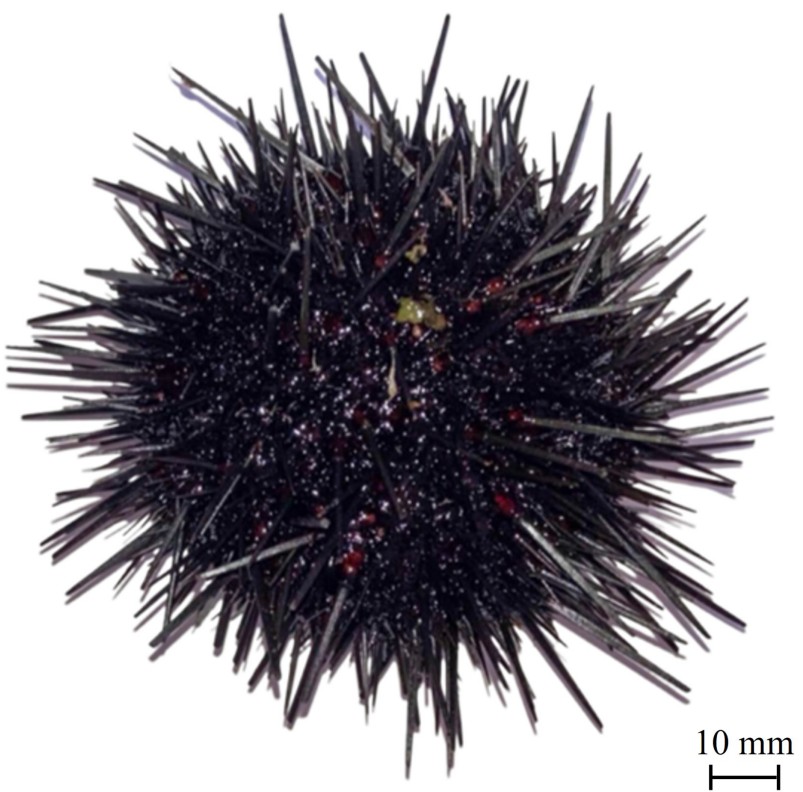

10 mm

**Figure 1** Sea urchin *Strongylocentrotus nudus*.

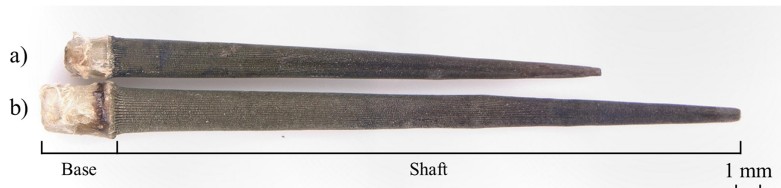

Base                                   Shaft                        1 mm

**Figure 2** Sea urchin spines in the (A) *ambulacral* and (B) *interambulacral* areas. Note: This image is for illustrative purposes only. All spines used for analysis were thoroughly cleaned prior to testing.

**Table 1 Dimensions of sea urchin spines on *Strongylocentrotus nudus* ($n = 10$ per area).**

|  | Total length (mm) | Base length (mm) | Shaft length (mm) | Diameter (mm) |
|---|---|---|---|---|
| Ambulacral area | 23.73 ± 2.45 | 2.08 ± 0.37 | 21.71 ± 2.15 | 0.52 ± 0.07 |
| Interambulacral area | 33.61 ± 3.97 | 2.88 ± 0.58 | 30.71 ± 3.40 | 0.59 ± 0.11 |

samples of 10 spines were observed using a digital microscope (VHX-5000; Keyence, Osaka, Japan). The transverse section area of the sea urchin spines, after subtracting the stereom structure, was measured for accurate calculations. The average percentage of the transverse sectional area was 80%. In addition, the bending modulus of elasticity ($E$),

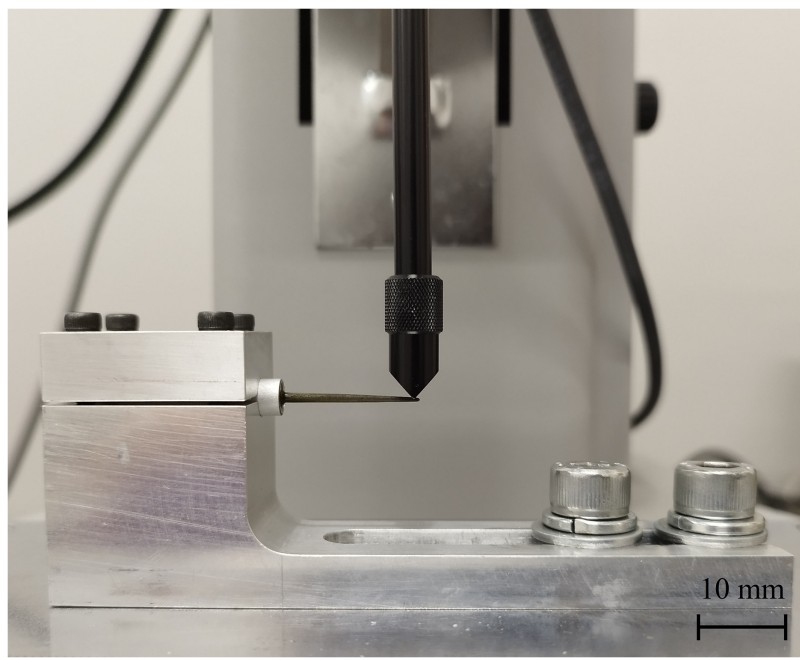

**Figure 3 Sea urchin spines fixed using a jig for the bending test.**

bending strength ($\sigma_B$), and maximum bending stress ($\sigma_{max}$) were determined using the following equations:

$$E = \frac{\sigma}{\varepsilon} = \frac{F/A}{\Delta L/L} \tag{1}$$

$$\sigma_B = \frac{F(L-x)}{Z} \tag{2}$$

$$\sigma_{max} = \frac{M_{max}}{Z} \tag{3}$$

where, $F$ is the maximum force, $A$ is the transverse section area, $\Delta L$ is the displacement, $L$ is the sea urchin spine length, $x$ is the spine position, $M_{max}$ is the maximum bending moment and $Z$ is the section modulus (*Moureaux et al., 2010*).

## Analysis using FT-IR

The spines were split into base and shaft parts and crushed for compound analysis. The size of the powdered spine fragments was between 40 and 100 µm. For analysis, the powdered samples of each part of the spine were prepared by placing approximately 1 to 5 mg on the plate. Calcite, dolomite, and magnesite in each section were analyzed using an FT-IR spectrometer (FT/IR 500; JASCO, Tokyo, Japan). FT-IR spectra were collected in an infrared reflection area of $30 \times 30$ µm$^2$ with a cumulative number of 1,024 times. The wavenumbers ranged from 800 to 1,500 cm$^{-1}$. Each region (ambulacral shaft, interambulacral shaft, and their bases) was measured three times, and representative spectra were averaged. For FT-IR analysis, powdered samples were prepared using

fragments from the entire shaft region to obtain a representative composition of the whole shaft.

## Evaluation of lattice spacing by XRD

To evaluate the lattice spacing by the standard silicon peak at $2\theta = 28.4°$ (*Wang et al., 2022*), powdered spine samples (0.2 g each) were prepared with 10% silicon powder for each part of the spine. The exact $2\theta$ of the calcite (104) and (006) planes in the spine was measured using an X-ray diffractometer (Rigaku Ultima IV, Tokyo, Japan). The diffractometer was operated at 40 kV and 40 mA at a $2\theta$ range of 27–33° with a step size of 0.01° (Cu-K$\alpha$ radiation: $\lambda = 1.5418$ Å). Then, the peaks (104) and (006) planes were determined by Lorentzian fitting software (OriginPro, Northampton, MA, USA).
The lattice spacing of the calcite phase in the spine was calculated from the $2\theta$ values of the spine's (104) and (006) diffraction peaks using Bragg's Law. The added silicon served solely as an internal standard to calibrate the instrument and verify measurement accuracy. The lattice spacing ($d$) of the spine was calculated using Bragg's law, as follows:

$$d = \frac{\lambda}{2 \sin \theta} \tag{4}$$

where $\lambda$ is the wavelength of X-ray radiation, and $\theta$ is the diffraction angle.

## Trace element analysis by ICP-AES

First, each powdered spine sample (0.1 g) was placed in an airtight container with 2 mL of nitric acid ($HNO_3$), and dissolved by heating at 100 °C for 40 min. Second, the dissolved samples were cooled to room temperature (20–25 °C), and pure water was added to make up a volume of 50 mL. The mixed samples were diluted 2,000-fold with pure water to the weight of Ca and 100-fold to the weight of Mg. For calibration, the standard solutions of Ca and Mg were diluted from 1,000 ppm to 5, 2.5, 1, 0.5, 0.25, 0.1, 0.05, 0.025, and 0.01 ppm. Finally, the trace elements were analyzed using ICP-AES (SPS3100HV UV; SII, Chiba, Japan). The samples of each part were measured using ICP-AES under conditions of high-frequency power (1.2 kW), and a cumulative number of 5 times. The concentrations of Ca and Mg in each part were measured, and the corresponding quantity (wt%) was calculated as follows:

$$wt\% = \left(\frac{M_C \times V \times D \times S_C}{M}\right) \times 100 \tag{5}$$

where, $M_C$ is the mass of the concentration, $V$ is the volume to be increased, $D$ is the dilution ratio, $S_C$ is the standard solution concentration, and $M$ is the mass of the sample.

## RESULTS

### Cantilever bending test

Figure 4 shows the force-displacement diagram of one representative spine per area and the distribution of bending failure load (N) for 10 spines per area for the cantilever bending test. The mechanical behavior of the fracture was linear, and the average maximum force

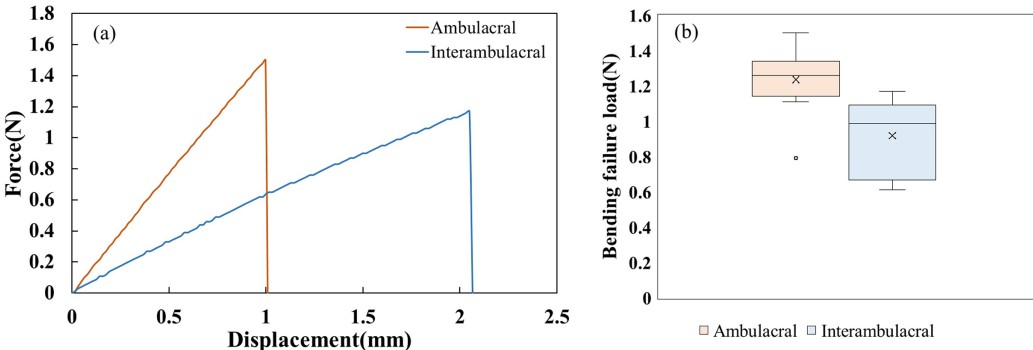

**Figure 4** (A) Representative bending force–displacement curves for ambulacral and interambulacral spines. (B) Distribution of bending failure load (N) for 10 spines per area. Each box shows the interquartile range (IQR), with the horizontal line representing the median and the "×" indicating the mean. Outliers are shown as individual points.

**Table 2** Mechanical properties of sea urchin spines.

|  | Ambulacral | Interambulacral |
| --- | --- | --- |
| Bending modulus (GPa), E | 52.067 | 10.133 |
| Bending strength (MPa), $\sigma_B$ | 631.75 | 300.71 |
| Maximum stress (MPa), $\sigma_{max}$ | 1,822.77 | 1,554.89 |

and displacement were determined from this relationship. Fractures occurred in the middle of three spines, in the tip of seven spines for the ambulacral area, and in the tip of 10 spines for the interambulacral areas. The average maximum force in the ambulacral spine was 1.2350 N, while in the interambulacral spine it was 0.9180 N. The average displacement in the ambulacral spine was 1.3265 mm, and in the interambulacral spine, it was 4.7523 mm. Table 2 lists the mechanical properties of spines in the ambulacral and interambulacral areas. The average bending modulus of spine elasticity in the ambulacral and interambulacral areas was 52.067 and 10.133 GPa, respectively. The average bending strength of the spine in the ambulacral area was 631.75 and 300.71 MPa in the interambulacral area. The average maximum stress of the spine in the ambulacral area was 1,822.77 and 1,554.89 MPa in the interambulacral area.

## FT-IR analysis

Figure 5 shows the FT-IR spectra of the (A) base and (B) shaft of sea urchin spines. Figure 5A shows that in the base of the spine in the ambulacral and interambulacral areas, calcite peaks appeared at 1,419 and 1,410 cm$^{-1}$, while magnesite peaks appeared at 855 and 861 cm$^{-1}$. Figure 5B shows the magnesite peak in the shaft part appeared at 860 and 855 cm$^{-1}$, while the calcite peak appeared at 1,418 and 1,420 cm$^{-1}$ (*Vecchio et al., 2007*; *Tanaka et al., 2019*). The dolomite peak occurred at 888 cm$^{-1}$ for both base and shaft part in both areas (*Bruckman & Wriessnig, 2013*).

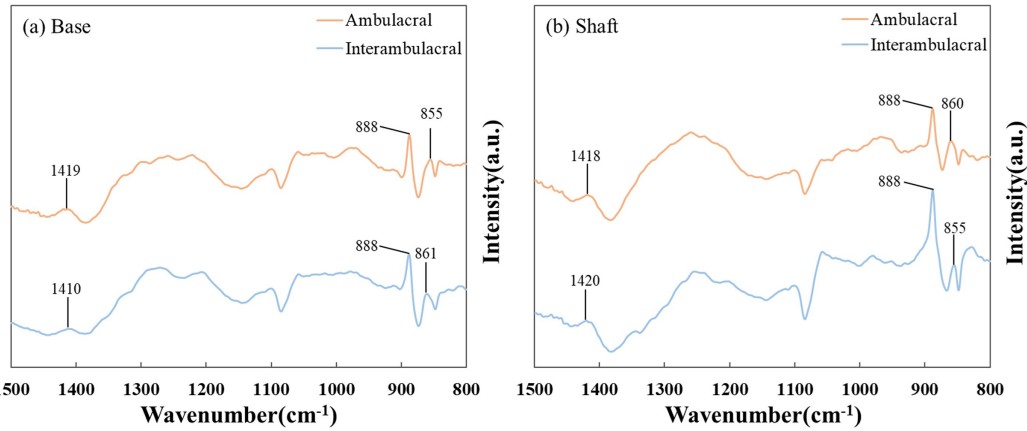

**Figure 5 Fourier-transform infrared (FT-IR) spectra of the (A) base and (B) shaft of sea urchin spines in the wavenumber range of 800–1,500 cm⁻¹.** To ensure reproducibility, each spectrum shown here is the average of three independent measurements conducted on separately prepared samples from each area.

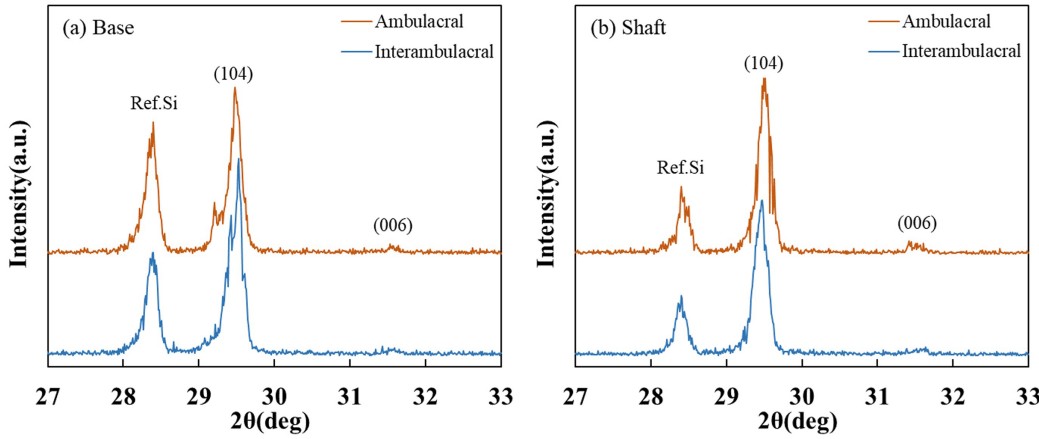

**Figure 6 X-ray diffraction (XRD) patterns of the sea urchin spine from (A) base and (B) shaft in the ambulacral and interambulacral areas.**

## Evaluation of lattice space using XRD

Figure 6 shows the XRD patterns of powdered sea urchin spine samples from each part of both areas. The peaks of the (104) plane appeared as follows (*Borzęcka-Prokop, Wesełucha-Birczyńska & Koszowska, 2007*): the base parts in the ambulacral and interambulacral areas were at 29.512° and 28.498°, respectively. The shaft angles in the ambulacral and interambulacral areas were 29.479° and 29.475°, respectively. The (006) peaks appeared as follows: the bases in the ambulacral and interambulacral areas were 31.581° and 31.563°, respectively. The shaft angles in the ambulacral and interambulacral areas were 31.529° and 31.544°, respectively. Figure 7 shows the lattice spacing of the sea urchin spines from each part in both areas. The lattice spacing increased from the base to the shaft in both areas. In the base part of the (104) plane, the lattice spacing in the ambulacral area was

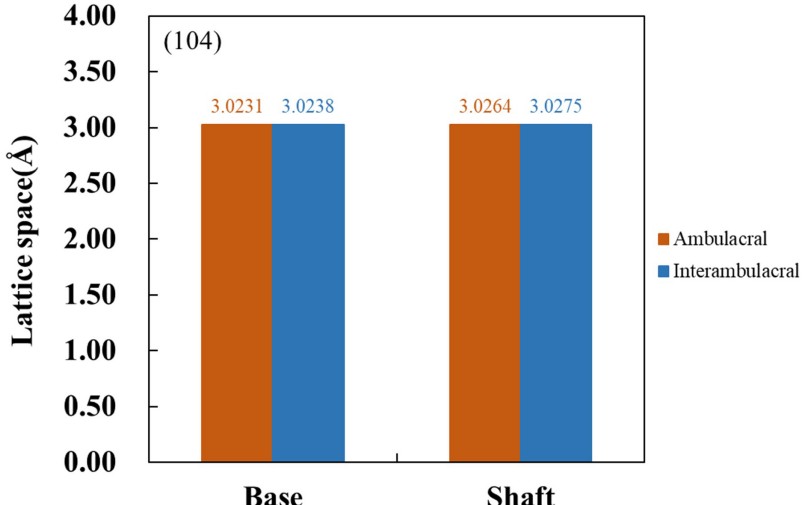

**Figure 7  Lattice space of (104) plane from the base and shaft in the ambulacral and interambulacral areas.**

narrower than that in the interambulacral area. The average lattice spacing of the (104) plane of the base part in the ambulacral and interambulacral areas were 3.0231 and 3.0238 Å, respectively. The average lattice spacing of the shaft part in the ambulacral and interambulacral areas were 3.0264 and 3.0275 Å, respectively.

## Trace element analysis

Figure 8 show the average concentrations (wt%) of Ca and Mg contained in each part of the spine from the ambulacral and interambulacral areas. In both areas, the Ca concentration in the base was lower than that of the shaft, while the Mg concentration was higher in the base compared to the shaft. In the base part of the spine, Ca concentration in the ambulacral area (35.4173%) was higher compared to the interambulacral area (31.6537%). The Mg concentration in the ambulacral area (1.1532%) was lower compared to that in the interambulacral area (1.2091%). In the shaft part of the spine, Ca concentration in the ambulacral area (39.6578%) was lower compared to that in the interambulacral area (42.1076%) while the Mg concentration was slightly higher in the ambulacral area (0.9866%) compared to that in the interambulacral area (0.9804%).

## DISCUSSION

In this study, we investigated the relationship between the mechanical properties and constituent elements in the ambulacral and interambulacral areas of *S. nudus* spines, focusing on their roles in locomotion and defense. Our analyses revealed differences in both the structural and compositional characteristics of the spines in the ambulacral and interambulacral areas. The spines in the ambulacral area demonstrated increased hardness and decreased elasticity, contributing to increased locomotory support by the tube feet. In contrast, the spines in the interambulacral area showed higher bending capacity and

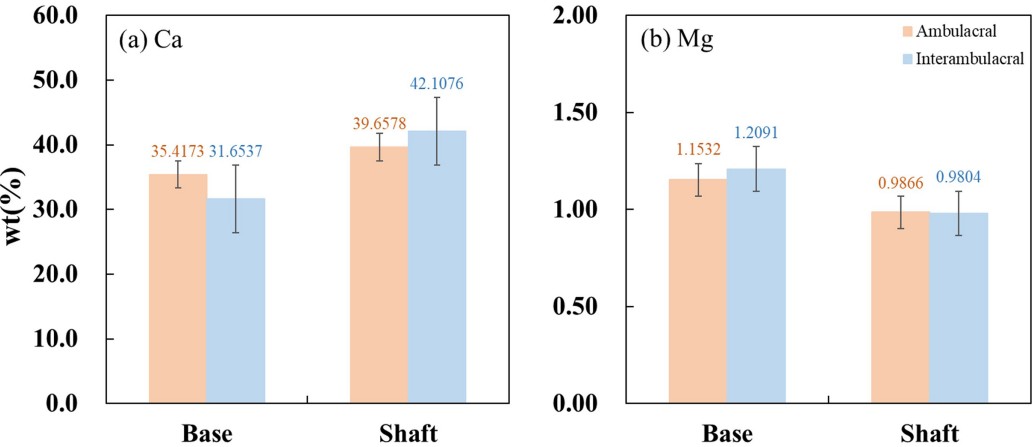

**Figure 8** Inductively coupled plasma atomic emission spectroscopy (ICP-AES) wt% of (A) calcium and (B) magnesium in the base and shaft in the ambulacral and interambulacral areas.

elasticity, a structure more suitable for defense through rapid outward extension in response to external stimuli (*Moureaux et al., 2010*; *Tsafnat et al., 2012*).

ICP and XRD analyses showed that the base of the spine generally contained more Mg than the shaft, suggesting that base strengthening is related to modifications in the crystalline lattice spacing in the (104) cleavage plane (*Deng, Jia & Li, 2022*). The lattice spacing in the ambulacral area was narrower than that in the interambulacral area, and the Mg concentration in the base of ambulacral spines was lower than in the interambulacral spines. However, the Mg concentration was higher in the shaft of the ambulacral area. The Mg distribution suggests functional specialization in the ambulacral area. The mechanical strength of the base, attributed to increased hardness from its crystalline structure, is crucial for protecting the organism (*Magdans & Gies, 2004*).

Damage to the base of the spine is often fatal, highlighting the importance of vital protection in this region (*Moureaux et al., 2010*; *Gorzelak et al., 2011*; *Albéric et al., 2019*). The increasing Mg concentration in the shaft is identified with narrower lattice spacing, referring to solid-solution strengthening where Mg displaces Ca, transforming calcium carbonate into magnesium carbonate and dolomite (*Lauer et al., 2020*; *Deng, Jia & Li, 2022*). This results in increased maximum bending stress, which may enhance performance during locomotion.

These structural and compositional variations may result from spatially controlled biomineralization. It is plausible that epithelial cells at different locations along the length of the spine regulate the secretion of organic matrix and ionic concentrations, particularly $Mg^{2+}$ and $Ca^{2+}$, during calcite crystallization. Such region-specific control may enable the formation of distinct crystal structures and mechanical properties tailored to localized functional demands.

Our results support the hypothesis that echinoid spines are structurally and functionally specialized based on the ambulacral and interambulacral areas, and in the base and shaft parts. The bending modulus observed in *S. nudus* spines (up to 52 GPa) is higher than that

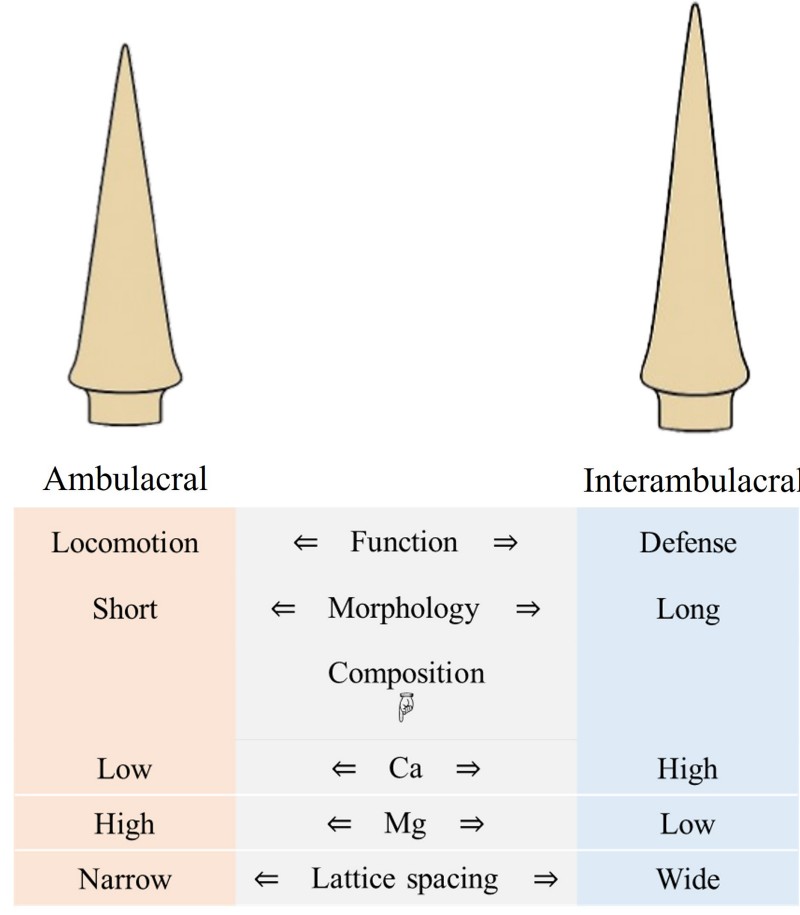

Figure 9 **Summary diagram comparing morphology, composition, and function of ambulacral and interambulacral spine shafts in *S. nudus*.** Note: The diagram focuses on shaft regions of the spines where most mechanical and compositional analyses were conducted.

of human cortical bone (up to 15–20 GPa) and approaches values reported for nacre (up to 60–70 GPa), suggesting high stiffness and resistance to deformation under mechanical loading. Previous studies have detailed stereom structure and mineral composition, beginning with foundational work by *Smith (1980)* and *Weber (1969)*, followed by more recent investigations (*Moureaux et al., 2010*; *Cölfen et al., 2022*), but our study is among the first to correlate microstructural differences with mechanical function between spine types. The observed variation in Mg concentration is consistent with previous findings, which links higher Mg levels to increased hardness in calcite (*Albéric et al., 2019*).

A schematic summary of these compositional, structural, and functional differences between ambulacral and interambulacral spines is provided in Fig. 9. This visual synthesis highlights the polarity in morphological features, elemental distribution, and mechanical behavior between spine types, reinforcing the observed functional specialization.

Controlling Mg concentration may function as a biological method to optimize mechanical properties by optimizing the hardness and elasticity. The orientation of

crystalline domains in interambulacral spines may also contribute to their controlled elasticity under mechanical stress. Understanding how Mg concentration and microstructure affect mechanical properties in sea urchin spines provides valuable insight into the design of bioinspired materials that combine transformability with strength (*Magdans & Gies, 2004*; *Moureaux et al., 2010*).

Although this study controlled for tip proximity by standardizing the indentation distance, future research could explore mechanical properties at equivalent distances from the spine base to isolate the effect of shaft length. Furthermore, high-resolution spatial mapping of elemental composition along the entire spine could reveal fine-scale biochemical variations, particularly near muscle or ligament attachment sites (*Vafidis et al., 2024*; *Baumiller, 2001*). Such spatially detailed analysis may deepen the understanding of functional adaptations in spine composition.

While our study is limited by analysis and may not fully capture living behavior, it nonetheless provides a foundational understanding of how microstructural and compositional factors, such as Mg concentration, affect echinoderm biomechanics. These findings offer a strong basis for future investigations. Future research should investigate environmental factors, such as seawater chemistry and hydrographic conditions, which have been shown to influence Mg:Ca ratios in echinoderm skeletons (*Roberta et al., 2020*; *Santos, da Gama & Flammang, 2018*; *Santos et al., 2023*), as well as the genetic regulation of the biomineralization mechanism. Moreover, investigating whether sea urchin spines are transformed in response to fracture or biological stress could reveal important transformative mechanisms. Comparative studies across different echinoid species may also provide valuable evolutionary insights into the functional specialization and morphological diversity of spines.

## CONCLUSIONS

This study provides an extensive analysis of the mechanical properties and constituent elements of sea urchin (*S. nudus*) spines, exhibiting substantial Mg concentration in the different functional areas between ambulacral and interambulacral areas. The higher Mg concentration in ambulacral spines results in increased hardness and narrow lattice spacing to improve support during locomotion. In contrast, the lower Mg concentration in interambulacral spines confers elasticity to resist cleavage fractures, optimizing them for defensive functions. These observations reveal how solid-solution strengthening and microstructural variations contribute to the functionality of a single biological structure (*Seto et al., 2012*; *Deng, Jia & Li, 2022*).

The specialized mechanical properties observed in sea urchin spines reveal important evolutionary adaptations and offer a compelling model for developing advanced biomaterials. Future research into atomic-level structure, compressive strength, and environmental influences on Mg concentration in sea urchin spine could further inform the design of bioinspired materials with tunable mechanical properties. These results have the potential to contribute to innovations in biomaterials and functional materials.

## ACKNOWLEDGEMENTS

We would like to thank N. Takeda, Instrumental Analysis Division, Global Facility Center, Creative Research Institution, Hokkaido University, for advice. We also thank Hashimoto, Kitami Institute of Technology, open facility center for analyzing sea urchin spines using inductively coupled plasma atomic emission spectroscopy, Fourier-transform infrared spectroscopy, and X-ray diffraction, to provide insights and expertise that greatly assisted this research. We are also grateful to Rishiri Umineko Guest House, who kindly provided the sea urchin samples used in this study.

### Funding

The authors received no funding for this work.

### Competing Interests

The authors declare that they have no competing interests.

### Author Contributions

- Pathitta Suteecharuwat conceived and designed the experiments, performed the experiments, analyzed the data, performed the computation work, prepared figures and/or tables, authored or reviewed drafts of the article, and approved the final draft.
- Mayuka Arakawa performed the experiments, analyzed the data, prepared figures and/or tables, and approved the final draft.
- Yutaka Yoshida conceived and designed the experiments, authored or reviewed drafts of the article, and approved the final draft.

### Data Availability

The raw data are available in the Supplemental Files.

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
