# Peer review of "Mechanical and chemical element structures of sea urchin spines: specialization in ambulacral and interambulacral areas"

_PeerJ Materials Science, doi:10.7717/peerj-matsci.38_

## Round 0.1 · original submission · Major Revisions

Reviewer 01 has some comments which are critical and need to be responded (as much as possible). Also in my opinion the Introduction (framing of the problem in context to the present knowledge) and Discussion needs severe improvement. Also the authors are requested to add a section (or a paragraph) describing the shortcomings of the study and future scopes [please note that objective evaluation of shortcomings will not have a negative impact on decision / acceptance of the article].

Reviewer 1 ·

Basic reporting

Paper by Suteecharuwat et al aims to show that magnesium concentration in the spines of sea urchin Strongylocentrotus nudus affects the mechanical properties of those spines.
The introduction doesn’t have a logical flow and could seem very chaotic to someone with limited knowledge on sea urchin skeleton. It is also not explained in what way “the mechanical properties of spines depend partly on their constituent elements” despite existing literature on the topic. Both discussion and introduction are missing vital papers about the role of magnesium in increasing mechanical strength of sea urchin biominerals (e.g. Wang et al, 1997 or Ma et al., 2008)
Some statements are also just not true, key example being: “the spine in the ambulacral area is used for walking, whereas that in the interambulacral area is used for defense.”, while noting that “currently, no study has elucidated the walking and defense functions of sea urchin spines” – and honestly, this paper does not make a case for either.

As for the English, there are plenty of wonky expressions or shortcuts like “this structure is expected to be useful in biomaterials” (“biomaterial development/research” would be more fitting), “Spine strength defends against fatal fractures” or some unfortunate expressions like “the bending characterization of a cantilever beam is particularly important for walking” (what does that even mean?).
I believe the manuscript would benefit from a review by a native English speaker to improve the language; a consultation with a zoologist to review specialistic terms would also be helpful.

Other reporting items: raw data provided; comments regarding abstract and title are below; some issues with figures listed at the end.

Title and abstract
- Title doesn’t seem to capture what the authors have been doing (i.e. finding the relationship between Mg content in the spine and the mechanical properties of the spine)
- throughout the manuscript: Strongylocentrotus nudus – species names should be written in italics (could be a PeerJ editor issue)
- “there was a significant difference in Mg concentractions…” – was a there a statistical test done? (if not, remove “significant” as it carries particular meaning)
- “the sea urchin acquired the necessary spinal function by controlling Mg concentration during walking” – very strangely sounding sentence

Introduction
Line 42 – better use “tests” rather than “shells” in the context of sea urchin anatomy (please correct throughout the manuscript)
Line 42 – incorrect citation. Did you mean Johnson et al, 2020? (https://www.sciencedirect.com/science/article/abs/pii/S0022098119301182?via%3Dihub)
Line 43 –citation missing from reference list (Kenneth et al. 2007)
Line 43 – should be Albéric et al., 2019 (not Albérica) (please correct throughout the manuscript)
Line 44 – a bit unfortunate phrasing, as it sounds like the shaft of the spine is a building block for stereom (not true). Stereom is a mesh-like microstructure found in all sea urchin ossicles, spines being an example (a neat explanation is given in Albéric et al., 2019, actually). A definition of stereom in the introduction will be helpful to the readers who are not familiar with zoological terms.
Line 47 – please provide examples with references (what constitutent elements and how do they affect mechanical properties of the spines?)
Line 47 – “this structure” – which structure? Stereom? It is not clear, looking at the preceding sentence.
Line 48 – “optimal for walking and defense”. Please provide relevant references.
Line 50 – “The bending characterization of a cantilever beam is particularly important for walking” – I am sorry, but this sentence is not very understandable. Do you mean that results of cantilever beam experiment could provide some information about walking? (far-fetched conclusion)
Line 54-56 – “The spine in the ambulacral area is used for walking, whereas that in the interambulacral area is used for defense.” – Where did this idea come from? Spines can support some movement, but bulk of the movement comes from tube feet contraction. Stating that function of a spine is related to interambulacral or ambulacral zone has no scientific backing – please provide references for such claims.
I do wonder if the authors somehow confused the above with primary and secondary spines, which might have different functions? (see introduction in Hebert et al. 2024 https://www.nature.com/articles/s41598-024-76239-7)

Materials & Methods
Line 68 – how many specimens and how many spines per specimen were analyzed? From line 80 we know that for cantilever bending test 20 spines were used (10/ area), what about other analyses?
Line 71 – was the organic tissue removed? If not, could it have affected the results of the bending test?
Line 81 – what was the maximum load?

Results
Line 143-145 – please check values and units as “interambulacral force” appears twice and has two different units
Line 146-148 – are those means? Maximum values? Medians?
Line 158 – correct citation Bruckman & Wriessnig 2013 (https://link.springer.com/article/10.1007/s10311-012-0380-4)
Line 162 – correct citation Borzęcka-Prokop et al. 2007 (https://www.sciencedirect.com/science/article/pii/S0022286006005369)
Line 176 – number of specimens? Was there any statistical analysis done?

Discussion
Discussion looks even shorter than introduction and has only four references. No mention of the mechanism for the differences between Mg concentration in the base and the shaft (e.g. spine ontogeny, see Moureaux et al. 2010). Similar results have been obtained by Magdans and Gies (2004), please cite this paper. I am also surprised as previous uses of bending tests in sea urchins are not really mentioned, as this could place the study in a wider context of the class.

General: Magnesium in sea urchins ossicles can be quite variable due to physiological effects (e.g. Smith et al, 2016, Iglikowska et al. 2017). How do your results place in the context of intraspecies Mg variability? (Have there been any other studies of S. nudus skeleton composition, considering it is a common Pacific species?) And importantly, have you considered doing a separate biomechanical analysis for open stereom and septa within the spines, considering septa can be rich in magnesium?

Line 194 – “In the ambulacral area, the hardness of the base is important for defense.” –Why would it only matter in the ambulacral area? Please provide references or remove completely.
Line 197-199 – citations for solid-solution strengthening as a mechanism for Mg incorporation. Also, last two sentences don’t read well.

Conclusions
Line 203 – the authors are placing a lot of confidence that “Solid-solution strengthening occurred in the spinal shaft.” That mechanism is not really well-explained, it is just assumed it happened, no references.

Table 1 – please include the number of specimens and spines (n=…) in the caption
Figures 8 and 9 – axis y should start at 0. Otherwise, any observed differences between bars seem larger and overstated. For Figure 9, what do the whiskers represents? SE, SD? Also, add information about sample size to the figure

Experimental design

See point 1.

Validity of the findings

Raw data provided, no statistics done for the paper

Reviewer 2 ·

Basic reporting

In the concerned article, the authors have tried to differential bio-materialistic and functional properties of sea urchin spines. Through an array of experimental and technical approaches, they properly differentiate between ambulacral and interambulacral areas of spines related to specified chemical deposition contributing to the elasticity, bending strength tensile nature.

Experimental design

The figures and experimental results are well-versed and coherent to the journal standard. The raw data supplied with the material were convincing and supportive to the result. These findings are handy in finding the better oriented biomaterials for various contemporary usage.

Validity of the findings

Although the findings are unique and could be implied in prospective considerations, there are few areas that were not clearly addressed. These are:
Major Comments:
1. Why did the authors chose this particular species of sea urchin? Biological feature of the species of interest is lacking which may be key orientation of this study choosing this particular species.
2. Is there other relevant data of likewise data available in other species. If so, additional furnishing of literature would be suitable. If not, a comparative study between two or more species would be more interesting to interpolate.
3. Whether the findings are species specific at greater extent.
4. Does these structural entities and chemical conditions related to the functional attributes of water vascular system/ambulacral system of sea urchin (being a member of Echinodermata). If so, then how the incurrent and excurrent water pressure related to the tensile strength of the spines, as suggestive in the manuscript, with special reference to ambulacral and interambulacral areas?
5. The discussion part lacks the interrelation between the findings and the physiological attributes of the sea-urchin apart from walking itself.

Minor comments:
1. There are some grammatical and structural ambiguities through-out the text which should be taken care of.
2. There are some typos that need to get rectified.

·

Basic reporting

How does the variation in Mg concentration affect the mechanical properties of sea urchin spines in different anatomical regions?
What is the relationship between the bending modulus of elasticity and the Mg concentration in the calcite structure of sea urchin spines?
How do the structural and compositional differences between the ambulacral and interambulacral areas influence spine function in sea urchins?
What role does the lattice spacing of calcite play in determining the mechanical strength of sea urchin spines?
How can the biomineralization strategy of sea urchins be applied to the development of novel functional biomaterials?

Experimental design

.

Validity of the findings

.

Additional comments

.

---

## Round 0.2 · Minor Revisions

Please respond to the reviewers comments.

Reviewer 4 ·

Basic reporting

Ln 53: Sea urchin tests are distinguished into ambulacral and interambulacral 54 areas (Gao et al., 2015). A very limited population of readers will know what this means. Please elaborate.
Ln 63: During locomotion, the spines support the body by providing mechanical stability and balance. Please reference
I see that secondary spines are not considered, nor mentioned herein. Why is that, and what do you expect their structural properties to be?
Ln 81: I note that the organic matrix of the spine was removed from the spines before measurement. How? Do you not want to measure the spine as a spine instead of as a crystal? Does the organic matrix not contribute to the physical properties of the spine? It would be important also to identify the contents of the organic matrix.
Many of the measurements of the physical properties of the spine will be opaque to the average reader. Can you add relationships that would make these number make sense to a broader audience?
It is not clear how lattice space was calculated. Please clarify.
Ln 227: Our results support the hypothesis that echinoid spines are structurally and functionally 228 specialized based on the ambulacral and interambulacral areas, and in the base and shaft parts. This is an important result/conclusion. It may have more impact as the/a title.
A diagram of result summary of the composition of spine and morphology and function would be very helpful.
Please speculate for the reader as to how these differences in composition might be constructed by the animal, especially along the length of the spine.

Experimental design

see above

Validity of the findings

see above

Additional comments

none

·

Basic reporting

In general, the manuscript is clearly structured and easy to follow. I did not notice major issues relating to grammar and style, but am obviously not a native speaker. I did, however, suggest some minor corrections in the annotated manuscript file, which is attached.

My major concern with the paper – as also pointed out by a reviewer of an earlier version of the manuscript is, that the authors failed to give sufficient credit to previous works on the topic and related studies. In many cases they fail to cite key contributions, but cite later, more specialized works instead. I do not suggest that the latter should be omitted, but, for example, when referring to echinoid stereom, I would strongly suggest to take a look at Smith 1980 (Smith, A.B. 1980 Stereom microstructure of the echinoid test. (Special Papers in Palaeontology). 25th Edition. – 81 pp., (The Palaeontological Association). Likewise, when referring to the chemical composition of echinoderm calcite I recommend also including the initial studies that set the grounds for latter works (e.g. Weber 1969, Weber et al. 1969).

Weber, J.N. (1969b) The incorporation of magnesium into the skeletal calcite of echinoderms. Am. J. Sci., 267, 537–566.

Weber, J., Greer, R., Voight, B., White, E. and Roy, R. (1969) Unusual strength properties of echinoderm calcite related to structure. J. Ultrastruct. Res., 26, 355–366.

A review paper that includes many of the references relevant for the present study has been published by James Nebelsick and myself a while ago: https://doi.org/10.1002/9781118398364.ch12

Experimental design

Having not applied most of the methods used in the present paper myself I will refrain myself to some basic comment regarding to the experimental design.

A) In Figure 2 it appears that the spines were not fully macerated before analysis and that some soft tissue was still adhering to the spines (and likely present in the stereom pores). This could potentially influence the measurements taken;

B) No information was given, how the skeletal elements were cleaned – in my experience enzyme solutions work best, followed by sodium hypochloride to fully remove tissue (mind that both methods need thorough rinsing afterwards); if this has indeed been done it needs to be stated in the material and methods section

C) Spines analysed were only classified into ambulacral spines and interambulacral spines. However, biologically the position along the oral-aboral axis of the animals may be much more relevant. Oral, ambital and adapical spine often fulfil different roles in sea urchins and in many species have even different shapes and sizes (and likely have differing chemical composition). For a meaningful comparison one would need to either restrict oneself to spines from a specific region on the corona or do a full comparison of all areas. In any case information on the position of the spines on the corona which were used in the analyses is missing.

D) Apparently, measurements were made in random rotational orientation of the spines. Given the observation that many echinoid spines are not perfectly rotationally symmetrical, but instead may show a slight aboral-oral differentiation and the three dimensionally and the nature of calcite with its oriented cleavage planes, which might affect response to directed load, I am wondering if the random orientations during measurements allow meaningful comparison between individual spines

E) FT-IR measurement: it is unclear where on the shaft the measurement was made

F) how many individuals were collected?

G) at which depth were they collected?

H) Please provide coordinates for the site

I) where are the specimens deposited?

J) Comparison of Mg-content of the spine base vs. the spine shaft possibly is to simplistic. Little information has been published as far as I am aware of, but there is some evidence that skeletal biochemistry (e.g. in echinoid teeth: Vafidis et al 2024: https://doi.org/10.1002/ece3.11251) and isotopic composition (Baumiller 2001. Light stable isotope geochemistry of the crinoid skeleton and its use in biology and paleobiology) may be variable in relation to the specific position within the respective skeletal element, depending on functional constraints such as muscle or ligament attachment sites. It might be useful to map Mg and Ca content in longitudinal cross sections of the spines.

Validity of the findings

It is difficult for me to comment on the validity of the findings given the issues raised above.

In addition, most figures present one datapoint for an ambulacral spine and one for an interambulacral spine only. I would expect to see some basic statistics – in Fig. 5, for example a second panel could be added showing all point of failures in one graph, so that the reader get some idea about the variance in the dataset. Likewise, how representative are the spectra shown in Fig. 6 – how many measurements were done – judging from the supplementary materials only a single measurement for each area and spine portion

I am also wondering if the differences observed could be an effect of spine length – the breakage test was performed always with the indenter positioned 1 mm below the tip of the respective spines, no matter how long they were (judging from the methods section) – I am curious if an analysis carried out at the same distance from the base of the spine would have given the same result.
It is also unclear if the authors checked the spines for signs of regeneration. Most echinoids are able to regenerate broken spine tips and shafts. Spine breakage and hence regeneration is very common in shallow water urchins like S. nudus. The regrown area has a different internal structure (lacking growth zones) in comparison to the non-regenerated part. This may affect the measurement too. Care should be to only use spines that show no signs of regeneration to avoid additional confounding factors.

Additional comments

The choice of the taxon studied should be explained in the manuscript not just in the reply to the reviewers.

The authors repeatedly use the term “shell” for the echinoid corona – this is incorrect – the echinoid skeleton is an internal skeleton, not an externally excreted shell

I am wondering how useful Fig. 4 is – in my opinion it does not add any information to the text and could be omitted.

Reference “Johnson et al., 2020” (line 44) is missing in the ref list

A number of recommendations by the reviewers of an earlier version of the manuscript were not considered in the revised version.

The following papers may also be relevant for the authors:
https://doi.org/10.1016/j.marpolbul.2023.115956
https://doi.org/10.3389/fmars.2018.00258 (see fig.3 for relationship between Mg-Ca rations and hydrographic variables)

---

## Round 0.3 · accepted · Accept

I find the revised manuscript suitable for publication, as the authors have adequately addressed all the reviewers' comments.